# Complexity-Aware Deep Symbolic Regression with Robust Risk-Seeking Policy Gradients

## Abstract

This paper proposes a novel deep symbolic regression approach to enhance the robustness and interpretability of data-driven mathematical expression discovery. Despite the success of the recent break-through, DSR, it is built on recurrent neural networks, purely guided by data fitness, and potentially meet tail barriers, which can zero out the policy gradient and cause inefficient model updates. To overcome these limitations, we use transformers in conjunction with breadth-first-search to improve the learning performance. We use Bayesian information criterion (BIC) as the reward function to explicitly account for the expression complexity and optimize the trade-off between interpretability and data fitness. We propose a modified risk-seeking policy that not only ensures the unbiasness of the gradient, but also removes the tail barriers, thus ensuring effective updates from top performers. Through a series of benchmarks and systematic experiments, we demonstrate the aforementioned advantages of our approach.

## 1 Introduction

Interpretability is an essential measure of machine learning models. Although large complex models, such as those with billions of parameters, have become standard practice in many fields, these models are typically black-box and provide little insight about the data. This can raise severe concerns regarding the reliability of deploying such models, particularly in scientific and engineering domains. Symbolic regression (SR) Jobin et al. (2019); Rudin (2019) is an important research direction to achieve interpretability in machine learning. Given a dataset that measures the input and output of a complex system of interest, symbolic regression aims to find a simple, concise equation to reveal the underlying mechanism of the system as to improve the understanding of the system and ensure reliability of the model.

For a long time, genetic programming (GP) Koza (1994); Randall et al. (2022); Burlacu et al. (2020) has been the mainstream approach for symbolic regression. However, GP is known to be computationally costly and time consuming due to the its evolutionary nature. The recent break-through, Deep Symbolic Regression (DSR) Petersen et al. (2019), instead uses a recurrent neural network (RNN) to generate expression trees, and employs a risk-seeking policy to train the RNN via reinforcement learning. While DSR is successful in many SR benchmarks, the RNN-based architecture might cause learning bottlenecks, such as vanishing gradients Hochreiter (1998), especially in large tree structures. In addition, DSR uses data fitness as the reward, which can tend to generate complex expressions to overfit the data, especially with the presence of noise. Furthermore, due to the usage of the reward difference as the weights in the policy gradients, DSR takes the risk of meeting tail barriers, namely, zero policy gradients, which can render the learning to be only driven by the entropy term, leading to over-exploration and inefficient model updates.

To overcome these limitations, we propose a novel approach, complexity-aware deep symbolic regression (CADSR). The major contributions of our work are summarized as follows.

- **Expression Model.** We use transformers Vaswani et al. (2023) to build the expression generation model, which can fully leverage the contextual information of each node in the expression tree and overcome RNN learning bottlenecks to improve the performance. To obtain a good representation, we design tree node encodings based on both the depth and horizontal position. We couple with breadth-first-search (BFS) to sample the expression tree, which is not only computationally efficient but also straightforward to implement.

- **Reward Design.** We design a Bayesian Information Criterion (BIC) as the reward function, for which we use the number of nodes and constant tokens in the expression tree to indicate the model complexity. BIC is justified by Bayesian model selection (Wasserman, 2000) and strongly connects to the minimum description length (MDL) (Rissanen, 1978), making it a principled and robust approach. In this way, our reward function considers not only the data fitness but also the expression complexity. The reinforcement learning can then seek to optimize the trade-off between the interpretability and data fitness of the expressions, avoiding producing over-complicated expressions that overfit the data, especially at present of noises.

- **Policy Gradient.** We propose a modified risk-seeking policy gradient for our BIC-based reward. Instead of using the reward difference as the weight to compute the gradient, we use a simple step reward mapping, which gives a constant weight. We show that this guarantees to prevent the learning from meeting any tail barrier or partial tail barrier. Accordingly, we can effectively leverage all the top performers for model updating, avoiding wild, less useful exploration.

- **Experiments.** We evaluated the performance of our method, CADSR, on the standard SR benchmark and ablation studies. In addition to DSR, we compared with seventeen other popular and/or state-of-the-art SR methods and several commonly used machine learning approaches. The performance of CADSR in both symbolic discovery and prediction accuracy is consistently among the best. In particular, the symbolic discovery rate of CADSR is the highest when data includes significant noises. In all the cases, CADSR generates the most interpretable expressions, while maintaining a high level of accuracy. CADSR outperforms the most comparable model, DSR, in all categories showing that it is a direct improvement. Extensive ablation studies further demonstrate the effectiveness of each component of our method.

## 2 Background

Given a set of input and output examples collected from the target system, denoted as $\mathcal{D} = \{(\mathbf{x}_i, y_i)\}_{i=1}^{N}$, symbolic regression aim to identify a concise mathematical expression that characterizes the input-output relationship, such as $y = \sin(2\pi x_1) + \cos(2\pi x_2)$. Deep symbolic regression (DSR) Petersen et al. (2019) discovers equations via an RNN-based reinforcement learning approach Sutton & Barto (2018), which can be broken down into four parts: environment, actor, reward, and policy.

The environment is designed to be the creation of an expression tree that represents a specific equation. Expression trees are directed trees where each node holds a token from the available list of operations and variables. For example, a common list of tokens would be $\{+, -, \times, /, \sin, \cos, x_1, 1\}$. Expression trees are built by selecting nodes in a preorder traversal of the tree with each tree starting, with only the root node. After a token for a node is selected, empty children will be added to node based on the token. Binary functions will have two empty children added, unary functions will only have one child, and variables will have no children. These trees are many-to-one mappings to the mathematical expressions, which causes an increase in the search space but prevents the generation of any invalid expression.

The actor is an RNN that predicts a categorical distribution of the available tokens for each node in the expression tree based on the hidden state of the RNN and the sibling and parent of the current node. Each token is randomly sampled from the categorical distribution. Additional rules are applied to the sampling process to prevent the selection of redundant operations or variables. Since the RNN builds the expression trees in a preorder traversal (POT) ordering, there can be a significant delay between the prediction of sibling tokens.

The reward function and policy drive the actor to explore and exploit the complex environment. In DSR, the reward function is a direct measurement of the data fitness of the generated expression,

$$R(\tau) = \frac{1}{1 + \text{NRMSE}}, \tag{1}$$

where $\tau$ denotes the expression, NRMSE represents the normalized root-mean-square error, and is defined as NRMSE $= \frac{1}{\sigma_y} \sqrt{\frac{1}{n} \sum_{i=1}^{n} (y_i - \tau(\mathbf{x}_i))^2}$ where $\sigma_y$ is the standard deviation of the training output in the dataset.

Once each expression tree has an associated reward, DSR applies a risk-seeking policy to update the actor according to its best predictions. Specifically, at each step, DSR samples a large batch of expressions, ranks their rewards, and selects the top $\alpha\%$ expressions to compute a policy gradient,

$$\nabla J_{\text{risk}}(\theta; \alpha) = \frac{1}{\alpha B} \sum_{i=1}^{B} [R(\tau^{(i)}) - R_\alpha] \cdot \mathbf{1}_{R(\tau^{(i)}) \geq R_\alpha} \nabla_\theta \log(p(\tau^{(i)}|\theta)), \tag{2}$$

where $B$ is the batch size, $\mathbf{1}_{(.)}$ is an indicator function, $\tau^{(i)}$ is the $i$-th expression in the batch, $\theta$ denotes the RNN parameters, $\log(p(\tau^{(i)}|\theta))$ is the probability of $\tau^{(i)}$ being sampled by the current RNN , $R_\alpha$ is the $1 - \alpha/100$ quantile of the rewards in the batch. Accordingly, all the equations below top $\alpha\%$ will not influence the update of the actor. Via such policy, the actor is able to generate worse-performing equations without it affecting the actor's overall performance — we only care about the top performed expressions. This allows for more unrestrained exploration and targeted exploitation of the top performers.

In addition to the policy gradient (2), DSR also introduces an "entropy bonus", to update the RNN actor parameters $\theta$. The entropy bonus is the average entropy gradient for the token distributions associated with the top $\alpha\%$ expression trees. Denote the entropy bonus as $\nabla J_{\text{entropy}}(\theta; \alpha)$, the RNN parameters $\theta$ are updated via

$$\theta \leftarrow \theta - l \cdot (\nabla J_{\text{risk}}(\theta; \alpha) + \nabla J_{\text{entropy}}(\theta; \alpha)), \tag{3}$$

where $l > 0$ is the learning rate.

## 3 Method

We now present CADSR, our new deep symbolic regression approach.

### 3.1 Transformer Actor

First, we design a transformer Vaswani et al. (2023) as the actor in the reinforcement learning framework. RNNs rely on a single hidden state that summarizes the information across all the previous nodes, which can be insufficient and is known to cause the vanishing gradient issue Hochreiter (1998) in larger structures. By contrast, the transformer explicitly integrate information of all the nodes to extract representations and make predictions, and hence can overcome the vanishing gradient issue and more effectively capture the nodes' relationships.

Specifically, to represent the expression tree nodes, we design a positional encoding (PE) to reflect their location information. Given a particular node, we consider both the depth, $d$, and horizontal position, $h$, in the tree. To align the horizontal positions of the nodes across different layers, we propose the following design. Denote the horizontal position of the parent node by $h_p$, if the node is the left child, we assign its horizontal position as $h = h_p - h_p/2^d$, otherwise we assign $h = h_p + h_p/2^d$. In this way, the parent node will be in between its two children horizontally, which naturally reflects the tree structure. The horizontal position for the root node is set to $1/2$. See Fig. 1 for an illustration. We develop a recursive algorithm to efficiently calculate the horizontal positions of all the nodes, as listed in Appendix Algorithm 3. Given $d$ and $h$, we then create a 2D-dimensional encoding,

$$\text{PE}(d, h, 2i) = \sin(\frac{d}{10000^{(4i/D)}}), \tag{4}$$

$$\text{PE}(d, h, 2i + 1) = \cos(\frac{d}{10000^{(4i/D)}}), \tag{5}$$

$$\text{PE}(d, h, 2j + D) = \sin(h), \tag{6}$$

$$\text{PE}(d, h, 2j + 1 + D) = \cos(h), \tag{7}$$

where $0 \leq i, j \leq \lfloor D/2 \rfloor$. The embedding of each tree node is the positional encoding plus the one-hot encodings of the tokens associated with the parent and sibling nodes (if these nodes exist). By default, every tree node — when created — is assigned an empty token.

To predict a token distribution for each node (according to which we can sample new tokens to generate expressions), we use multiple self-attention layers across all the nodes in the tree, followed by a linear layer, and then apply softmax to the output to produce the token distribution for each node.

**Expression Sampling.** To generate an expression, we need to dynamically grow the expression tree and sequentially sample the token for each node, since we do not know the tree size and structure beforehand. DSR uses a preorder traversal to sample the token for each node. Since the order indexes of the nodes can vary with tree growth, one needs to re-vectorize the whole tree at every step, and hence it can be inefficient. To overcome this problem, we use the breadth-first-search (BFS) order. Since the tree is expanding layer by layer, the ordering of the previous nodes will never change, and therefore we only need to perform vecotrization once before performing any sampling. This is not only efficient but straightforward to implement. Given the token distribution, we first fill out invalid tokens and then sample from the remaining. Whenever the tokens have formed an valid expression, we stop the sampling and return the expression. The expression tree sampling is summarized in Algorithm 2 of the Appendix.

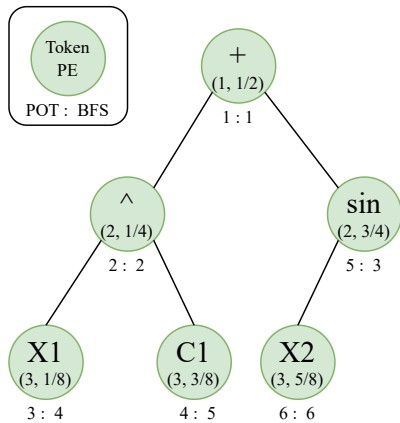

Figure 1: Expression tree for $y = x_1^{c_1} + \sin(x_2)$. POT and BFS denote the node order for preorder traversal and breadth-first-search, respectively.

### 3.2 BIC Reward Function

Interpretability is a key motivation for symbolic regression. However, if one uses data fitness as the reward to guide the learning, like in DSR, it will tend to learn lengthy, complex expressions to overfit the data, especially when data contains noise, which is common in practical applications. This can hurt the interpretability of the learned expressions. To address this problem, we use Bayesian information criterion (BIC) (Schwarz, 1978) to construct a new reward function. As a principled and robust approach, BIC is derived by maximizing the model evidence through integrating out the model parameters, and is thus justified by Bayesian model selection (Wasserman, 2000). BIC has a known similarity to minimum description length (MDL) Rissanen (1978), allowing BIC to be interpreted as an approximation of MDL.

To reflect the complexity of a sampled expression, we set $k$ to the number of nodes in the expression tree plus the number of constant tokens. The inclusion of the number of constant tokens is twofold: first, since the values of the constant tokens are unknown apriori — we need to estimate their values from data, they actually sever as unknown model parameters (like those in neural networks) and introduce extra degrees of freedoms, which increase the model complexity. Second, this can also prevent the actor from selecting an unnecessary number of constant tokens, which can cause a significant increase in runtime for optimizing their values. Our BIC reward is given below,

$$\text{BIC}(\tau) = k \log(N) - 2 \log(L(\tau)) \tag{8}$$

where $\log(L(\tau)) = \sum_{i=1}^{N} \log \mathcal{N}(y_i | \tau(\mathbf{x}_i), \sigma^2)$ is a (log) Gaussian likelihood, $\sigma^2$ is the variance of the training outputs, and N denotes the number of training points.

### 3.3 Robust Risk-Seeking Policy

While our BIC reward can account for expression complexity, applying it to the risk-seeking policy presents several challenges. The first challenge is that, because BIC is unbounded, directly incorporating it into equation (2) compromises unbiasedness.

**Lemma 3.1.** *When using the BIC reward* (8)*, the risk-seeking policy gradient* (2) *is not guaranteed to be unbiased.*

---
**Algorithm 1** Complexity-Aware Deep Symbolic Regression (CADSR)

---
**input** Learning rate $l$; risk factor $\alpha$; batch size $B$; coefficients $\lambda > 0$ and $\lambda_{\mathcal{H}} > 0$
**output** The best equation $\tau^*$
 1: Initialize transformer with parameters $\theta$
 2: **while** $i < epochs$ **do**
 3:     $\mathcal{T} \leftarrow \{\tau^{(i)} \sim p(\cdot|\theta)\}_{i=1}^B$ {Sample $B$ expressions using the transformer}
 4:     $\mathcal{T} \leftarrow \{\text{OptimizeConstants}(\tau^{(i)})\}_{i=1}^N$ {Optimize the values of the constant tokens}
 5:     $\mathcal{R} \leftarrow \{\text{BIC}(\tau^{(i)})\}_{i=1}^N$ {Calculate the reward for each expression using (8)}
 6:     $\mathcal{R}_\alpha \leftarrow (1 - \alpha/100)$-quantile of $\mathcal{R}$ {Find the reward that denotes the top $\alpha\%$}
 7:     $\mathcal{T} \leftarrow \{\tau^{(i)} : \mathcal{R}(\tau^{(i)} \geq \mathcal{R}_\alpha\}$ {Reduce the sampled expressions to the top performers}
 8:     $g_1 \leftarrow \frac{\lambda}{\alpha B} \sum_{i=1}^{\alpha B} \nabla_\theta \log p(\mathcal{T}|\theta)$ {Compute for top $\alpha\%$ gradient}
 9:     $g_2 \leftarrow \frac{\lambda_{\mathcal{H}}}{\alpha B} \sum_{i=1}^{\alpha B} \nabla_\theta \mathcal{H}(\mathcal{T}|\theta)$ {Compute entropy gradients}
10:     $\theta \leftarrow \theta + l(g_1 + g_2)$
11:     **if** $\max \mathcal{R} > \mathcal{R}(\tau^*)$ **then** $\tau^* \leftarrow \tau^{(\arg\max \mathcal{R})}$ {Update best equation}
12: **end while**
13: **return** $\tau^*$

---

We leave the analysis in Appendix Section D.1. To overcome this problem, a commonly used strategy, which is also adopted in DSR, is to introduce a continuous mapping that maps the reward value to a bounded domain. For example, DSR uses the mapping $f(z) = \frac{1}{1+z}$ where $z \in [0, \infty]$, which maps the unbounded NRMSE (in $[0, \infty]$) to the range $(0, 1]$; see (1). Another choice can be the sigmoid function, $s(z) = \frac{1}{1+e^{-z}}$ that maps from $(-\infty, \infty)$ to $(0, 1)$. In doing so, we can apply Leibniz rule to interchange integration and differentiation to show the unbiasedness of the policy gradient. However, this strategy brings up the second challenge. That is, the reinforcement learning can encounter a *tail barrier*, defined as follows.

**Definition 3.2** (Tail barrier). Let $\alpha \in [0, 1)$. A risk seeking policy meets a $\alpha$-tail barrier if the top $\alpha\%$ rewards of the sampled actions (*e.g.*, expression trees) are all equal to $R_\alpha$.

**Lemma 3.3.** *Given any continuous mapping $f$ that can map unbounded reward function values to a bounded domain (e.g., $(-\infty, \infty) \rightarrow [0, 1]$), suppose the reward function is continuous, there always exists a set of distinct rewards values that numerically create a tail barrier in the risk-seeking policy.*

The proof is given in Appendix Section D.2. In practice, since we often use continuous reward functions (*e.g.*, NRMSE or Gaussian likelihood) and reward mappings, there is a risk of encountering the tail barrier. From (2), we can see that the tail barrier can incur *zero* policy gradient, since all the top $\alpha\%$ rewards are identical to $R_\alpha$, leading to a zero weight for every gradient. As a consequence, the actor model would not have any effective updates according to the feedback from the selected expressions (top performers). In DSR, the RNN model will be updated only from the entropy bonus (see (3)), and henceforth the learning starts to explore wildly.

To address these challenges, we use a step function to perform a reward mapping,

$$f(z) = \begin{cases} \lambda, & \text{if } z > R_\alpha \\ 0, & \text{otherwise} \end{cases} \tag{9}$$

where $\lambda > 0$ is a constant. Then the risk-seeking policy gradient is given by

$$\nabla J_{\text{risk}}(\theta; \alpha) \approx \frac{\lambda}{\alpha B} \sum_{i=1}^B \mathbf{1}_{\text{BIC}(\tau^{(i)}) > R_\alpha(\theta)} \nabla_\theta \log(p(\tau^{(i)}|\theta). \tag{10}$$

**Lemma 3.4.** *By using the step function (9) for reward mapping, the policy gradient with our BIC reward, as shown in (10), is unbiased and will not encounter any tail barrier.*

The proof is given in Appendix Section D.3. Our approach is summarized in Algorithm 1.

## 4 Related Work

DSR is a reinforcement learning method that uses an RNN actor Petersen et al. (2019). The risk-seeking policy rewards the actor only for top-performing expressions, which allows the actor to make a diverse range of predictions without being punished for poor average performance.

Many symbolic regression models have started to use transformers, such as Kamienny et al. (2022); Valipour et al. (2021); Vastl et al. (2022). However, all of these models learn the mapping directly from the dataset to the expression trees in a supervised fashion. This type of models is named as *Deep Generative Symbolic Regression (DGSR)* by Kamienny et al. (2023). To achieve this end-to-end learning, a much larger model architecture is necessary, along with a substantial collection of datasets. By contrast, DSR and our method are not DGSR; they essentially perform unsupervised learning, and the dataset only corresponds to one (unknown) expression. Hence, the model size and data quantity is much smaller. Recently, Shojaee et al. (2023) developed Monte Carlo tree search (MCTS) with transformers (TPSR) to conduct DGSR. TPSR assumes a *pre-trained* transformer is given, and conducts an efficient search to identify well-performed expressions. TPSR introduces a regularization term to control the expression complexity. The reward is specified as $r(\tau(\cdot)|\mathbf{x}, y) = \frac{1}{1+\text{NMSE}(y,\tau(\mathbf{x}))} + \lambda \exp(-\frac{l(\tau(\cdot))}{L})$ where $l(\tilde{f}(\cdot))$ retrieves the number of tokens in the given expression, and $\lambda > 0$ is the weight of the regularization term. Note that our work does *not* assume the availability of a pre-trained model. Throughout the evaluation, our method has never utilized a pre-trained transformer. Nonetheless, our ablation study demonstrates that our framework can be easily adapted to incorporate pre-trained models and gain their benefits; see Section 5.3 Kamienny et al. (2023) proposed DGSR-MCTS, which also uses MCTS search but uses a critic network — a transformer based model — to scores expressions without using the reward function, which allows for incomplete expressions to be evaluated by the critic.

Another relevant recent work is symbolic physics learner (SPL) (Sun et al., 2023), which uses MCTS but employs a different regularized reward function, $r(\tau(\cdot)|\mathbf{x}, y) = \frac{\eta^n}{1+\sqrt{\frac{1}{N}\sum_{i=1}^{N}||y_i-\tau(\mathbf{x}_i)||_2^2}}$, that has a discount factor $\eta$ that is raised to the $n$, which represents the number of product rules in the expression tree. This regularized reward function selects equations that use the fewest product rules, but does not have any regularization based on the length of the expression.

Lastly, the recent work uDSR Landajuela et al. (2022) combines several different methods together to improve DSR. One of the additions is a transformer trained using supervised learning or reinforcement learning to provide the RNN actor with additional information about the dataset. uDSR also incorporates genetic programming to each expression tree generation step, which allows for a larger variety of expression trees to be generated each epoch. Validating the performance of the different parts of uDSR required several ablation studies, where every combination of methods, including DSR, was tested in the uDSR paper Landajuela et al. (2022). The addition of a transformer to DSR showed minimal improvement and occasionally hindrance to uDSR during the ablation study. Furthermore, genetic programming was the single biggest influence on performance, showing that there needs to be an improvement to DSR. uDSR appeared on the Pareto frontier near the optimal mixture of accuracy and complexity.

## 5 Numerical Experiments

For evaluation, we examined CADSR in the well known comprehensive SRBench dataset La Cava et al. (2021), and then we studied each component of CADSR to confirm their efficacy.

### 5.1 Overall Performance

In SRBench, we first tested on the 133 problems with known solutions. We ran eight trials for each problem at the four *noise levels*, 0%, 0.1%, 1%, and 10%. We ran CADSR in a large computer cluster, for which we set a time limit of 6 hours for each trial. We deployed the trials on 10 V100 GPUs with parallelization of 6 processes per GPU. The maximum number of epochs is set to 2000 without early termination. The architecture of the transformer is shown in Appendix Fig. 8. In each epoch, we sampled a batch of 1000 expressions to compute the policy gradient. We used ADAM optimization where the learning rate was set to 5E-4. The full list of hyperparameters of our method is provided in Appendix Table 1. We compared with 17 popular and/or state-of-the-art SR methods

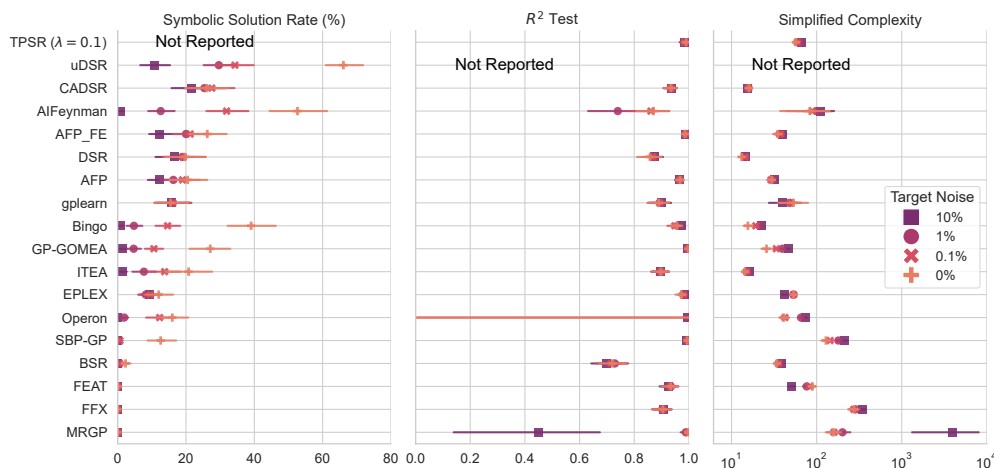

Figure 2: Symbolic regression performance on 133 problems from SRBench with known solutions.

in terms of Symbolic Solution Rate (%), $R^2$ test, and Simplified Complexity, which are standard metrics for SR evaluation. These SR baselines include AIFeynman Udrescu & Tegmark (2020), AFP_FE Schmidt & Lipson (2009), DSR Petersen et al. (2019), Bingo Randall et al. (2022), etc. The majority of these are genetic programming methods with a few notable exceptions: DSR is deep reinforcement learning, BSR Jin et al. (2020) is an MCMC method with a prior placed on the tree structure, and AIFeynmen is a divide-and-conquer method that breaks the problem apart by hyper-planes and fits with polynomials. The results of the competing methods are retrieved from the public SRBench report La Cava et al. (2021). We show the results of all the methods in Fig. 2.

Overall, CADSR shows strong performance in symbolic discovery as measured by Symbolic Solute Rate. In particular, when data includes significant noise (10%), CADSR achieves the best solution rate, showing that our method is more robust to noise than all the competing methods. Meanwhile, the simplified complexity of our discovered expressions is among the lowest. This together shows that our method, with the BIC reward design, not only can find simpler and hence more interpretable expressions, but also is more resistant to data noise. The $R^2$ Test shows the prediction accuracy of the discovered expressions. As we can see, the $R^2$ Test obtained by CADSR is close to the best. The slightly better methods, such as AFP_FE and GP-GOMEA, however, generate lengthier and more complex expressions, which lack interpretability and are much far away from the ground truth expression. It is worth noting that CADSR outperforms DSR in both Symbolic Solution Rate (%) and $R^2$ Test, showing an improvement on both expression discovery and prediction accuracy. When data does not have noise,

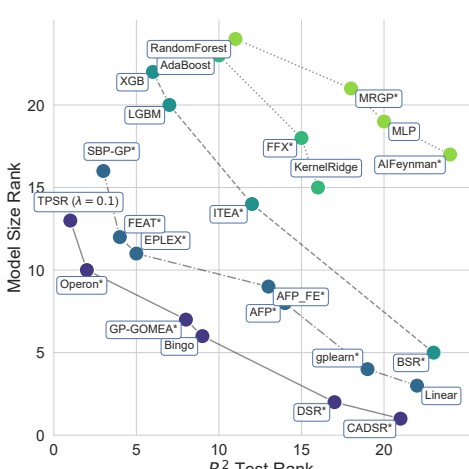

Figure 3: Pareto front of each method in 120 black-box problems of SRBench; the true solutions are unknown.

AIFeynman, uDSR, and Bingo shows better Symbolic Solution Rate than CADSR. This might be because AIFeynman tends to use polynomials to construct the expressions, which match most of the ground-truth; Bingo as a genetic programming approach, uses evolution operators to sample new expressions, which might explore more broadly; uDSR is an ensemble approach using AIFeynman, genetic programming, and DSR and thus can achieve higher symbolic accuracy.

Next, we tested with the 120 black-box problems in SRBench. Since the true solutions for these problems are *unknown*, we examined the Pareto front of all the methods in Model Size Rank *vs.* $R^2$ Test Rank. We also retrieved the results of several commonly used machine learning algorithms from

the SRBench report La Cava et al. (2021) for comparison. These algorithms include Random Forest, AdaBoost, and Multi Layer Peceptron (MLP). The setting of these methods are given in La Cava et al. (2021). As shown in Fig. 3, CADSR is at the most frontier, meaning CADSR is among the best in terms of the trade-off between the model size (expression complexity) and $R^2$ test (prediction accuracy). It is interesting to see that among those most frontier methods, CADSR tends to find the most concise expressions (ranked as the lowest in model size) while sacrificing the prediction accuracy to a certain degree. On the contrary, methods like Operon, typically generates way more complex expressions yet with smaller prediction error. It shows that our method can push the best trade-off toward more interpretability, which can be important in practical applications.

## 5.2 Ablation Study

Next, we performed ablation studies to confirm the efficacy of each component of our method. We selected a subset of problems from SR Bench that shows the most significant difference between DSR and CADSR. These problems include: `Feynman_I_12_11` and `Strogatz_sherflow1` that demonstrate the biggest decrease in $R^2$ score and symbolic solution rate, `Feynman_test_9` showing the increase in $R^2$ score, and `Feynman_I_34_27` and `Feynman_III_12_43` exhibiting the largest increase in symbolic accuracy. For each problem, we ran five trials at $10\%$ noise with each ablation using the same seed. Below is a brief description of each ablation.

**Transformer Actor**. We first tested altering the RNN to a transformer while maintaining the remaining DSR framework, including original prediction ordering. DSR does not have a defined positional encoding, so we used a standard 1D-dimensional positional encoding based on the position of the token in the prediction order, *i.e.,* pre-order traversal (POT). This comparison would be between DSR and DSR-POT-1PE in Figures 4, 5, and 6, showing that switching from an RNN to a transformer in the DSR framework incurs some trade-off in learning. This change is highlighted by the difference in symbolic discovery rate between the problems, as DSR with a transformer can consistently solve problems that the RNN can not, and vice versa. Since this is a curated selection of problems to highlight the difference between DSR and CADSR, we have magnified the difference between the two methods, causing this stark difference in performance.

**Prediction Ordering**. Next, we validated the change of the prediction ordering from the pre-order traversal (POT) to a Breadth First Search (BFS) ordering while maintaining the 1D-dimensional position encoding. DSR-POT-1PE and DSR-BFS-1PE compare these two tests in Figures 4, 5, and 6. The improvement from the prediction ordering change can be easily seen in Figures 4 and 5, as DSR-BFS-1PE has better $R^2$ scores and symbolic accuracy. Furthermore, we can decrease the model's runtime from 0.00361 to 0.00157 seconds per equation by changing the prediction ordering because BFS enables the model to keep token generation on the GPU during equation sampling.

**Position Encoding**. To confirm the effectiveness of our designed 2-dimensional PE, we tested with alternatives, including 1D-dimensional PE based on the tokens' location in the sequence of the BFS order. These two tests are DSR-BFS-1PE and DSR-BFS-2PE in Figures 4, 5, and 6. The 2D-dimensional PE improved $R^2$ scores and consistency of $R^2$ scores while maintaining a similar symbolic discovery rate for the test problems.

**Policy**. Next, we evaluated the effectiveness of our robust risk-seeking policy while using the NMSE-based reward function. DSR-BFS-2PE denotes the risk-seeking policy test, while the robust risk-seeking policy test is CADSR-NMSE. Figures 4 and 5 show that the new policy actually hinders the model's performance in both $R^2$ score and symbolic accuracy. The most likely cause of this is the lack of convergence from the robust risk-seeking policy, as it relies on the top $\alpha\%$ of expression to guide better equations rather than a numerical gradient. However, when the robust risk-seeking policy is combined with the BIC reward function it was designed for, we can see a an improvement in symbolic accuracy and $R^2$ score. A proof of the requirement of the robust risk-seeking reward is provided in Appendix D.3 and additional analysis in actual learning processes is done in Appendix C.2. These differences in policy performance could be alleviated by using a nonconstant gradient with a robust risk-seeking gradient, but this is left for further research.

**Complexity vs. Fittness Reward.** Lastly, we tested the BIC reward function compared to NMSE, and two newer regularized reward functions introduces with TPSR and SPL, as specified in (4) and (4), respectively. These four tests are labeled CADSR-BIC, CADSR-NMSE, CADSR-SPL, and CADSR-TPSR. Figures 4 and 5 show that BIC can consistently improve the model's performance in

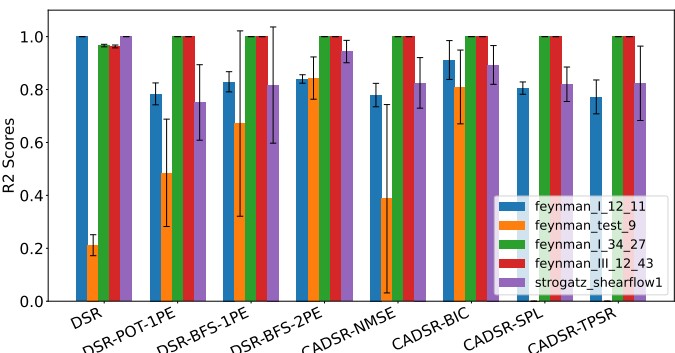

Figure 4: $R^2$ scores across various ablations of CADSR, namely, prediction ordering (POT vs. BFS), 1D or 2D position encoding (1PE vs. 2PE), policy (DSR vs. CADSR), and reward function (NMSE, BIC, SPL, TPSR). The comparison spans five problems with the most significant differences between DSR and CADSR. This comparison highlights the improvement of the DSR-BFS-2PE and shows that BIC is the best reward function with CADSR.

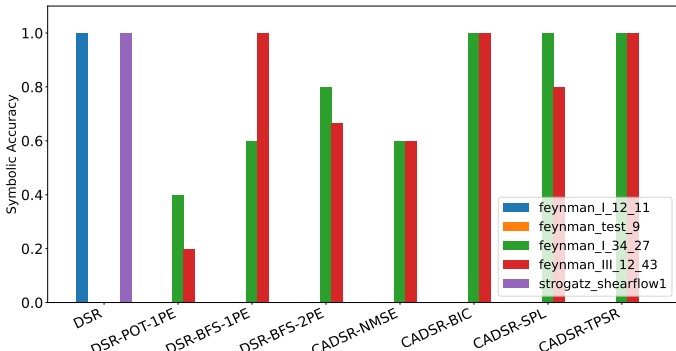

Figure 5: Symbolic accuracy across various ablations of CADSR, namely, prediction ordering (POT vs. BFS), 1D or 2D position encoding (1PE vs. 2PE), policy (DSR vs. CADSR), and reward function (NMSE, BIC, TPSR). The comparison spans five problems with the most significant differences between DSR and CADSR. This figure highlights the significance of a regularized reward function (BIC, SPL, TPSR) as they have the highest symbolic recovery rate. Additionally, the figure shows that the switch from an RNN to a transformers incurs some trade-off in learning (DSR vs. all other ablations).

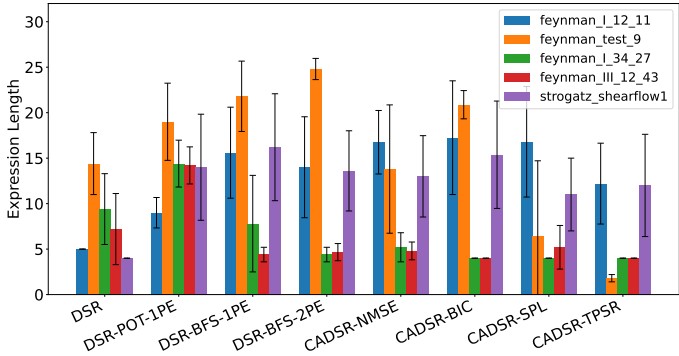

Figure 6: Expression length across various ablations of CADSR, namely, prediction ordering (POT vs. BFS), 1D or 2D position encoding (1PE vs. 2PE), policy (DSR vs. CADSR), and reward function (NMSE, BIC, SPL, TPSR). The comparison spans five problems with the most significant differences between DSR and CADSR. This figure highlights the impact of the regularized reward function in shortening expression length and improving the convergence consistency of the expressions.

Figure 7: Pre-trained CADSR symbolic accuracy on three problems from SR Bench.

| Dataset | Expression |
|---|---|
| Feynman_I_12_4 | $\frac{q_1 r}{4\pi\epsilon r^3}$ |
| Feynman_I_18_12 | $rF\sin(\theta)$ |
| Strogatz_Glider_1 | $-0.05x^2 - \sin(y)$ |

(b)

both $R^2$ and symbolic accuracy when compared to using the NMSE reward function. Furthermore, we see from Figure 6 that BIC helps reduce the expression length while maintaining $R^2$ compared to the other reward functions. BIC is the best-regularized reward function for our method when compared to SPL and TPSR's reward functions. Both other regularized reward functions hinder CADSR accuracy or symbolic discovery rate by over-selecting for shorter functions. The best example is the relative performance on Feynman_test_9, where SPL and TPSR's reward functions prevent any notable symbolic discovery causing the $R^2 \leq 0.0$ ( Figure 4) because of the high pressure towards short equations (Figure 6). An examination of BIC improving learning, and shortening expressions is provide in Appendix C.1.

### 5.3 Pre-Trained Transformer

Although pre-training is not the focus of CADSR, we wanted to see if CADSR has the potential to be improved through pre-training. One of the benefits of a transformer is pre-training, which can be used to add information about the entire system. We generated a random dataset from pre-training by randomly sampling expression trees; a more in-depth explanation is left in Appendix E. In order to do pre-training, we need to add information about the problem into the sequence, which we do by adding additional channels into the sequence to contain the $(\mathbf{x}, y)$ information. We pre-trained the method using risk-seeking reinforcement learning to maintain the same environment that CADSR runs in. We tested the pre-trained method on two problems from SRBench that have a dimensionality of 3 and $0\%$ noise to match the pre-training dataset. Lastly, we leveraged the transformer's log scaling rule by increasing the transformer's size to reduce training time Kaplan et al. (2020).

Overall, the pre-trained version of CADSR drastically improved equations that were similar to the training set. We can see that the Feynman_I_12_4 has worse performance, which is most likely due to the fact the training dataset has a large number of sin, cos, log, and exp functions as they are equally likely to be sampled, whereas in most cases they are uncommon. This change in performance is further highlighted by the performance of the other two equations. Overall, we can see that pre-training has the potential to be highly beneficial to this method and is left for further research.

### 5.4 Runtime

We conducted a runtime comparison between gplearn, DSR, and CADSR and found that CADSR had comparable runtimes at 0.00157 seconds per equation as compared to 0.00105 seconds per equation for gplearn, and 0.0014 seconds per equation for DSR. Furthermore, we found an that the BFS ordering reduced runtime by 50% and that the pre-trained CADSR increases runtime by nearly 1500%. The exact details of the study are given in Appendix F.

## 6 Conclusion

We have presented CADSR, a new symbolic regression approach based on reinforcement learning. On standard SR benchmark problems, CADSR shows promising performance. The ablation study confirms the effectiveness of each component of our method. Nonetheless, our current work has two limitations. First, the implementation is inefficient, especially for the expression sampling coupled with validation rules. This leads to a slow training process. Second, we lack an early stopping mechanism to further reduce the training cost and prevent useless exploration. In the future, we plan to address these limitations and use our method in more practical applications.

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

## A  Algorithms

Below, we show the algorithms for expression sampling and positional encoding generation. Note that for expression sampling, we over-sample expressions so that we can return a high number of unique ones. If not enough unique expressions exist, then we begin to allow duplicates to fill out our batch size requirements.

## B  Model Details

Table 1 and Fig. 8 show the comprehensive hyperparameter settings and the architecture of the transformer used in our method. Note that we use the Levenberg–Marquardt algorithm to optimize constant tokens for each discovered equation.

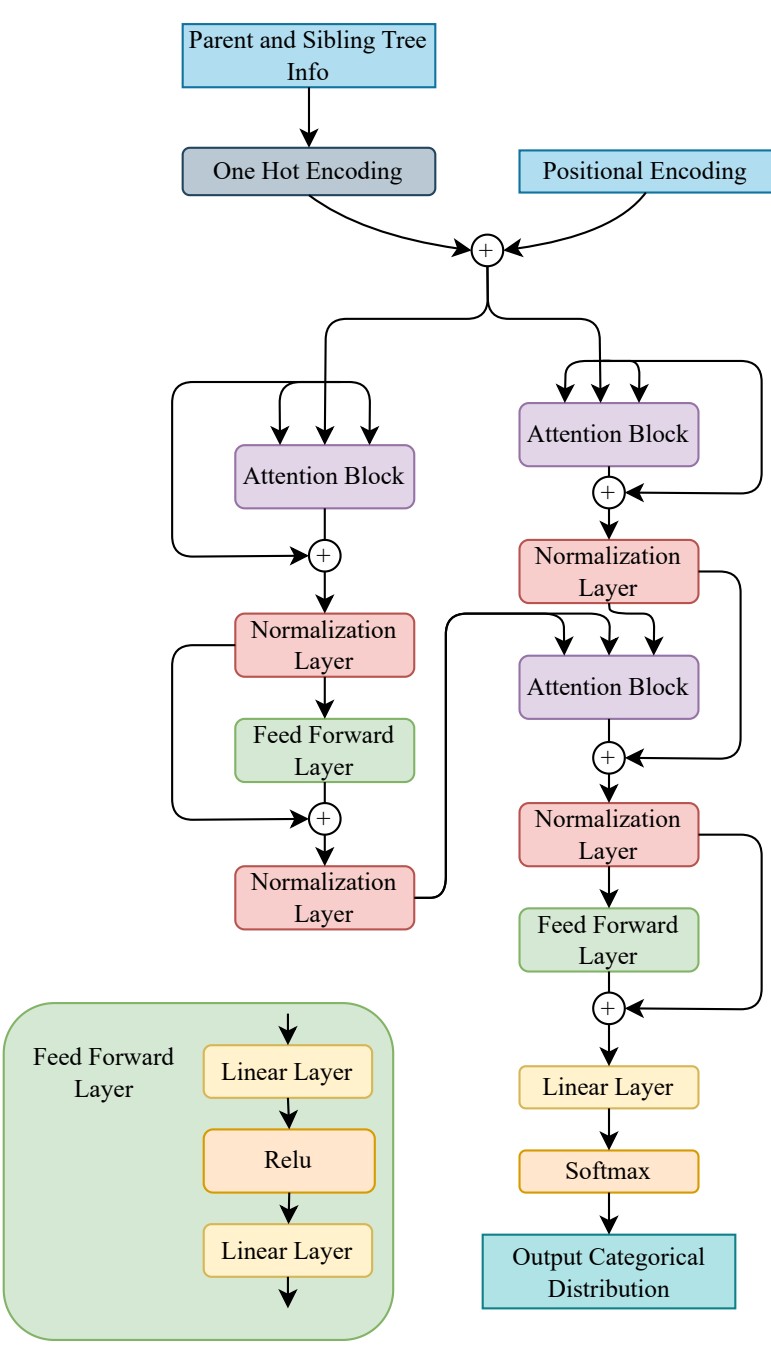

Figure 8: The architecture of the transformer actor in CADSR.

---

**Algorithm 2** Expression Tree Sampling

---

**input** Number of expressions to sample $B$; oversampling scalar $\gamma > 1$, maximum tree-node number $\nu$
**output** A set of expressions $\mathcal{T}$

 1: $\mathcal{T} \leftarrow$ ExpressionTrees($\gamma B$) {Creates $\gamma B$ empty expression trees}
 2: **while** $i < \nu$ **do**
 3:     $V_{\mathcal{T}} \leftarrow$ Inputs($\mathcal{T}$) {Fetching the input embeddings of all the expression trees}
 4:     $S \leftarrow p(V_{\mathcal{T}}|\theta)$ {Predicting categorical distributions from the transformer}
 5:     $S \leftarrow R(S)$ {Applying rules to each distribution}
 6:     $K \leftarrow P(\cdot|S)$ {Sampling from the categorical distribution to obtain tokens}
 7:     $\mathcal{T}_i \leftarrow K$ {Adding the new tokens into the expression trees}
 8: **end while**
 9: $\mathcal{T} =$ Unique($\mathcal{T}, B$) {Take the first $B$ Unique expression trees}
10: **return** $\mathcal{T}$

---

**Algorithm 3** Positional Encoding Generation

---

**input** An expression tree $\tau$

 1: $\tau$.root_node.depth = 1
 2: $\tau$.root_node.horizontal = 1/2
 3: PositionEncodingInformation($\tau$.root_node)

---

**Algorithm 4** PositionEncodingInformation

---

**input** Current node

 1: **if** node has left **then**
 2:     node.left.depth = node.depth + 1
 3:     node.left.horizontal = node.horizontal - $1/(2^{\text{node.left.depth}})$
 4:     PositionEncodingInformation(node.left)
 5: **end if**
 6: **if** node has right **then**
 7:     node.right.depth = node.depth + 1
 8:     node.right.horizontal = node.horizontal + $1/(2^{\text{node.right.depth}})$
 9:     PositionEncodingInformation(node.right)
10: **end if**

---

Table 1: Hyperparameter settings of CADSR.

| *Hyperparameter* | *CADSR* | *Pre-Trained* |
|---|---|---|
| Variables | $\{1, c \text{ (Constant Token)}, x_i\}$ | $\{1, c \text{ (Constant Token)}, x_1, x_2, x_3\}$ |
| Unary Functions | $\{\sin, \cos, \log, \sqrt{(\cdot)}, \exp\}$ | $\{\sin, \cos, \log, \sqrt{(\cdot)}, \exp\}$ |
| Binary Functions | $\{+, -, *, /, \hat{\ }\}$ | $\{+, -, *, /, \hat{\ }\}$ |
| Batch Size | 1000 | 1000 |
| Risk Seeking Percent ($\alpha$) | 0.05 | 0.05 |
| Learning Rate | 5E-4 | 5E-4 |
| Max Depth | 32 | 32 |
| Oversampling | 1.5 (Ideally 3) | 1 (Ideally 3) |
| Number of Epochs | 2000 | 2000 |
| Policy | BIC | BIC |
| Entropy Coefficient $\lambda_{\mathcal{H}}$ | 0.005 | 0.005 |
| Policy Coefficient $\lambda$ | 0.04 | 0.04 |
| Encoder Number | 1 | 4 |
| Decoder Number | 1 | 4 |
| Number of Heads | 1 | 4 |
| Feed Forward Layers Size | 2048 | 2048 |

Table 2: Hyperparameter settings for DSR

| Hyperparameter | DSR |
|---|---|
| Batch Size | 100 |
| Learning Rate | 0.0005 |
| Entropy coefficient | 0.005 |
| Risk Factor | 0.05 |
| RNN Type | LSTM |
| Layer Number | 1 |

| Label | Epoch | $R^2$ | Expression |
|---|---|---|---|
| 1 | 0 | 0.9771 | $((x_1 * (((x_1^{c_0})/(\sqrt{(c_1)}))) * c_2)) * \ldots$ |
| 2 | 2 | 0.9999 | $(c_0 - (\sqrt{((c_1 - (c_2 * ((((((c_3 * x_0) \ldots$ |
| 3 | 14 | 1.0 | $(\log(((c_0^{(((((((x_1/x_0)/(x_1/c_1))^{c_2}} \ldots$ |
| 4 | 33 | 1.0 | $(((x_1 * (c_0^{c_1})) * ((\cos(c_2)) * (((( \sqrt{(c_3)}) + 1) - \ldots$ |
| 5 | 36 | 1.0 | $((x_1 * ((x_0 + (x_1 - c_0))/c_1)) * c_2)$ |
| 6 | 242 | 1.0 | $c_0 x_1 - x_0 x_1 - x_1^2$ |

(a)

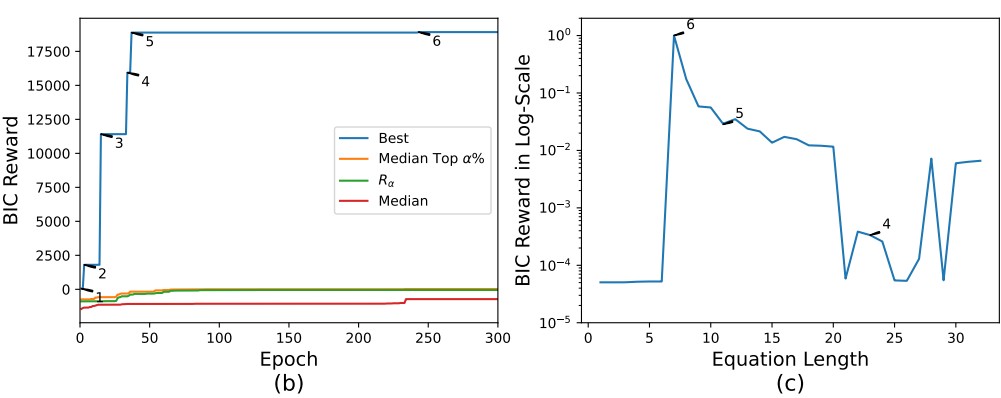

(b)        (c)

Figure 9: Learning progress with BIC reward function on *Strogatz_lv2* problem where the true solution is $y = 2x_1 - x_0 x_1 - x_1^2$

## C Additional Ablation Analysis

### C.1 BIC Reward Analysis

On the other hand, the lengthy expressions found in Table **??** indicates that the DSR reward (1), which is purely based on data fitness, does not seek to reduce model complexity to enhance the interpretability. We further replace the DSR reward with our BIC reward and re-run the learning process. While across five runs, the discovery rate and $R^2$ test scores remain 100% and 1.0, respectively, the discovered expressions are much simpler and closer to the true solution; see Table 3 of the Appendix. In Fig. 9, we further showcased the learning progress of one trial with our BIC reward function. It can be seen clearly from Fig. 9 that along with more epochs, the learned expressions are increasingly concise and finally exactly recover the true solution. Meanwhile, from Fig. 9b, we can see that those increasingly simpler expressions receive increasingly bigger rewards, while Fig. 9c further confirms that shorter expressions do receive larger rewards. All these demonstrate that our BIC reward indeed guides the learning toward more concise and, hence, more interpretable expressions in addition to fitting the data well.

| Trial | Full Expression | $R^2$ | Symbolically Accurate |
|---|---|---|---|
| 1 | $((2.0 - x_0) - x_1)x_1$ | 1.0 | Yes |
| 2 | $(2.0 - (x_0 + x_1))x_1$ | 1.0 | Yes |
| 3 | $(x_1 * (3.0 - (((x_0/x_0) + x_0) + x_1)))$ | 1.0 | Yes |
| 4 | $((1 - ((x_1 + x_0) + -1)) * x_1)$ | 1.0 | Yes |
| 6 | $(((\log(7.389)) - x_0) - x_1)x_1$ | 1.0 | Yes |

Table 3: Trials of using BIC reward on *Strogatz_lv2* problem, where the true solution is $y = 2x_1 - x_0 x_1 - x_1^2$.

## C.2 Tail Barrier Analysis

Finally, we examined the learning behavior with our robust risk-seeking policy. To this end, we tested on the *Strogatz predprey1* problem in SRBench. We ran our method for one trial and then replaced our policy with the risk-seeking policy as used in DSR (see (2)) to run another trial. The random seeds for the two runs were set to be the same. In Fig. 10a and 11a, we report how the best reward (among the sampled expressions at each step), the median reward, $R_\alpha$, and the median of the top $\alpha\%$ rewards (denoted as Median Top $\alpha\%$), varied along with the training epochs. Obviously, when the median of the top $\alpha\%$ rewards are identical to $R_\alpha$, many weights (around 50%) in the risk-seeking policy (2) become zero (since $R(\tau^{(i)}) = R_\alpha$), resulting zero gradients for the corresponding expressions. This can be viewed a "partial" tail barrier. To examine how many top $\alpha\%$ expressions are pruned due to their gradients being roughly zeroed out, we used a threshold $10^{-3}$ and report the portion of such expressions at each epoch in Fig. 10b and 11b. From Fig. 10a, we can see during the running of original risk-seeking policy, we encountered three significant events at epochs 250, 975, and 1750. In these events, the median of the top $\alpha\%$ rewards almost overlap with $R_\alpha$, causing a large portion of the top $\alpha\%$ expressions pruned due to that their gradients were nearly zeroed out. Accordingly, the learning afterwards can be more dominated by entropy bonus (see (3)) rather than those best-performed expressions. By contrast, while during the course of our method, there are also quite a few events when the median of the top $\alpha\%$ almost overlap with $R_\alpha$, no expression in the top $\alpha\%$ have their gradients zeroed out, since every such expression is assigned a constant reward $\lambda$; see (9) and (10). Therefore, the update of the model parameters will always leverage the information from all top $\alpha\%$ expressions, and hence can effectively avoid over-exploration.

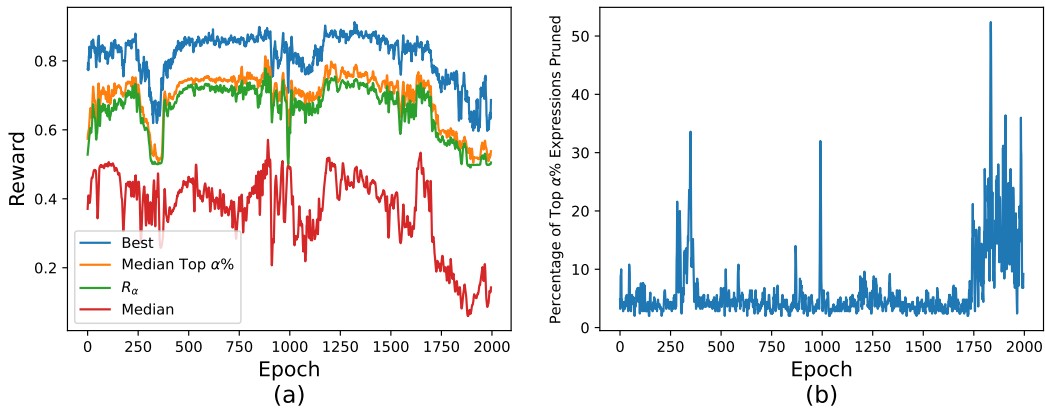

Figure 10: Running CADSR with risk-seeking policy (2) on *Strogatz predprey1* problem. The top-performed expressions can be pruned from being used for model updating, due to nearly zeroed out gradients.

# D Theoretical Results

## D.1 Proof of Lemma 3.1

*Proof.* The value of the BIC reward function is essentially a random variable, since it is determined by the random expression sampled by the transformer actor. Let us denote this random variable by $Z$ and its probability density by $p(z|\theta)$ where $\theta$ denotes the transformer parameters. Obviously,

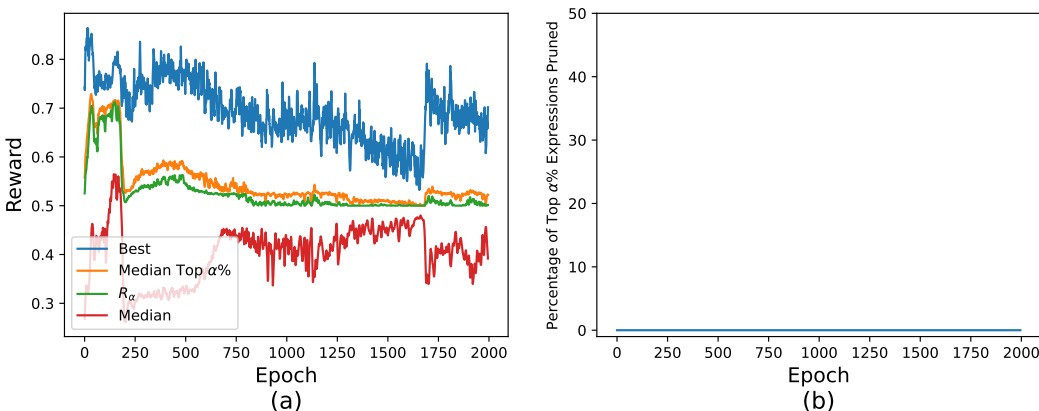

Figure 11: Running CADSR with robust risk-seeking policy (10) on *Strogatz predprey1* problem.

$Z \in (-\infty, \infty)$. The risk-seeking policy aims to approximate the gradient of the expectation over the truncated random variable $Z_\alpha = Z \cdot \mathbf{1}_{Z \geq s_\alpha}$, where $s_\alpha$ is the $1 - \frac{\alpha}{100}$ quantile of the distribution of $Z$,

$$s_\alpha = \inf\{z : \mathrm{CDF}(z) \geq 1 - \frac{\alpha}{100}\}. \tag{11}$$

The probability density of $Z_\alpha$ is given by

$$p(z_\alpha|\theta) = \frac{1}{\alpha} p(z|\theta) \mathbf{1}_{z \geq s_\alpha}. \tag{12}$$

The gradient of the expectation is computed as

$$\mathbb{E}[Z_\alpha] = \frac{1}{\alpha} \int_{s_\alpha}^{\infty} z p(z|\theta) \mathrm{d}z, \tag{13}$$

$$\nabla_\theta \mathbb{E}[Z_\alpha] = \frac{1}{\alpha} \cdot \nabla_\theta \int_{s_\alpha}^{\infty} z p(z|\theta) \mathrm{d}z. \tag{14}$$

Since the integration upper-bound is $\infty$, we cannot apply Leibnize rule to equivalently interchange the gradient and the integration. In other words, if we switch the order of the expectation (integration) and the differentiation, the result can be changed. Therefore, the policy gradient in the form of (2) is no longer guaranteed to be unbiased.

$\square$

### D.2 Proof of Lemma 3.3

*Proof.* Pick up any reward value $z_0$. Due to the continuity of the mapping $f$, for arbitrary $\epsilon > 0$, there exists $\delta > 0$ such that for all $z \neq z_0$, if $|z - z_0| < \delta$, then $|f(z) - f(z_0)| < \epsilon$. Let us take $\epsilon = \frac{1}{2}s$, where $s$ is the machine precision (*e.g.*, $2^{-32}$). Since the reward function is continuous, we can find a set of distinct reward values $z_1, \ldots, z_M$ from $B(z_0, \delta(\epsilon)) = \{z \in \mathrm{dom}\, f, |z - z_0| < \delta(\epsilon)\}$. Let us look at the mapped rewards, $f(z_1), \ldots, f(z_M)$. For any $1 \leq i, j \leq M$, we have

$$|f(z_i) - f(z_j)| = |f(z_i) - f(z_0) + f(z_0) - f(z_j)| \leq |f(z_i) - f(z_0)| + |f(z_0) - f(z_j)| < \frac{s}{2} + \frac{s}{2} = s.$$

Therefore, there are no numerical difference among these mapped rewards, and they can create a tail barrier.

$\square$

### D.3 Proof of Lemma 3.4

*Proof.* For any set of BIC reward values, $\mathcal{S} = \{R(\tau^{(1)}), \ldots, R(\tau^{(B)})\}$, we denote the mapped reward values by $\widehat{\mathcal{S}} = \{\widehat{R}_1, \ldots, \widehat{R}_B\}$, where each $\widehat{R}_j = f(R(\tau^{(j)}))$ ($1 \leq j \leq B$). We know that each

$$\widehat{R}_j = \begin{cases} 1 & R(\tau^{(j)}) > R_\alpha \\ 0 & R(\tau^{(j)}) \leq R_\alpha \end{cases} \tag{15}$$

where $R_\alpha$ is the $1 - \frac{\alpha}{100}$ quantile of the rewards in $\mathcal{S}$. Therefore, the $1 - \frac{\alpha}{100}$ quantile of the mapped reward values $\widehat{\mathcal{S}}$, namely, $\widehat{R}_\alpha = 0$. Since all the top $\alpha\%$ rewards in $\widehat{\mathcal{S}}$ take the value $\lambda$, which is strictly bigger than $\widehat{R}_\alpha = 0$, the tail barrier in the risk-seeking policy will never appear. Furthermore, every expression in the top $\alpha\%$ will not have their gradient zeroed out (see (2)), since the weight of the gradient is the constant $\lambda$ (see (10)). That means, we even will not meet a partial tail barrier.

To show the unbiasedness, we need to replace $R_\alpha$ by the $1 - \alpha/100$ quantile of the distribution of the BIC reward $R$. Denote the mapped reward by $\widehat{R}$. Since $\widehat{R}$ is bounded, namely, $\widehat{R} \in [0, 1]$, we can follow exactly the same steps in the proof of the original risk-adverse gradient paper Tamar et al. (2014) to show the unbiasedness of (10). Note that, with the step mapping $f$, the expectation, $\mathbb{E}[\widehat{R}\mathbf{1}_{\widehat{R} > Q_\alpha}]$ where $Q_\alpha$ is the $1 - \alpha/100$ quantile of the distribution of $\widehat{R}$, is actually a constant, namely $\lambda\alpha$. Hence the gradient of the expectation is zero. That means, our new risk-seeking policy gradient (10) is asymptotically converging to zero (with $B \to \infty$). However, in practice, $B$ is always finite, and our policy gradient is rarely close to zero. Rather, it will leverage all the top performers to conduct efficient model updates.

$\square$

## E Pre-trained Transformer

### E.1 Dataset Generation

In order to generate the dataset to train the transformer on, we randomly generated expression trees and sampled multiple constants for each expression tree while maintain uniqueness of the expression tree skeleton. Each expression tree was randomly generated by sampling an uniform policy in a breadth first search ordering. Since the uniform policy gave each token equal probability, the generated expressions had a higher number of unary functions than typical of the Feynman problems. For each expression, a random set of $\mathbf{x}$ was sampled from a uniform distribution on the interval $(-5, 5)$ and used to compute $y = \tau(\mathbf{x})$. We stored the breadth first search ordering, the positions, and the $(\mathbf{x}, y)$ to train the transformer on. The dimensionality of the problems to three for this feasibility test. The final dataset contained $\sim 80,000$ expressions.

### E.2 Training

The model was trained in the same reinforcement learning framework that CADSR use for a single problem with an additional 250 points attached to the sequence. The additional 250 points was selected as a balance between information, and increased compute time. We found this training method to work best as it allowed for the transformer to learn the relationship between the dataset and expression trees without covering towards a single expression. The model was trained for $\sim 5000$ epochs on an RTX 3080 over the course of 2 days.

Alternatively, we tried training the model in a supervised learning framework. This training method failed to work well, as during testing the model converged very rapidly to a finite number of equations. This failure is not indicative of supervised learning pre-training with CADSR being impossible, but needing more than a naive implementation.

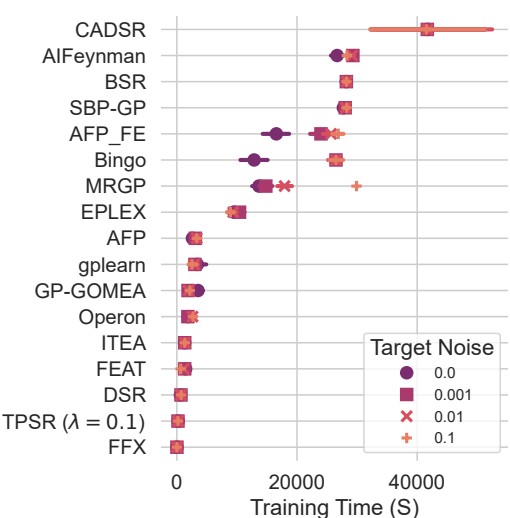

Figure 12: Reported time comparison by SRBench

## F   Run Times

Overall, it is difficult to compare runtimes, as CADSR primarily use the GPU, while gplearn, and DSR primarily use the CPU. As such an improvement to either the GPU or CPU hardware would alter the runtime comparison. In Figure 12 we provide the recorded times by SRBench, however this chart is not a very accurate comparison as CADSR does not have an early stopping criterion and shared resources for this benchmark.

| Method | Full Runtime (s) | Runtime Per Equation (s) |
|---|---|---|
| DSR | 116 | 0.0014 |
| gplearn | 42 | 0.00105 |
| Preordering | 7211 | 0.00361 |
| CADSR | 3135 | 0.00157 |
| Pre-trained CADSR | 42524 | 0.0213 |

Table 4: Run time comparison on Nguyen-2 on a Titan V GPU with an 32 Intel Xeon Silver 4108 CPU

