# OpenReview forum: "Complexity-Aware Deep Symbolic Regression with Robust Risk-Seeking Policy Gradients"
_ICLR.cc/2025/Conference — ICLR 2025 Conference Withdrawn Submission_

### Official Review · Reviewer_s71y · 2024-11-01

**Soundness:** 2
**Presentation:** 3
**Contribution:** 3
**Rating:** 5
**Confidence:** 4

**Summary:**

In this work, the authors build on DSR, a well-known reinforcement-based algorithm for Symbolic Regression. Specifically, they replace the original reward function with the Bayesian Information Criterion (BIC) to create a balance between interpretability and data fitness. Additionally, instead of using an RNN, they employ a transformer model coupled with breadth-first search (BFS) and custom positional embeddings to mitigate the vanishing gradient problems associated with RNNs. Finally, instead of computing the gradient using the reward difference, they utilize step reward mapping to avoid tail barriers. Through experiments on well-established benchmarks, the authors demonstrate that their method yields the most interpretable expressions while maintaining a high level of accuracy.

**Strengths:**

- The paper is clearly written and easy to follow
- The topic is relevant—interpretability is at the core of symbolic regression.
- The results are interesting, especially in the experiments involving noise. Specifically, the proposed method achieve the highest accuracy compared to all other baselines on SRBench in the 10% noise setting, making this method not only interpretable but also quite robust.

**Weaknesses:**

- My main concern is that the proposed method seems to require significantly more computational resources than other approaches (one order of magnitude more compute than DSR, i.e., 116 vs. 3135 as shown in Table 4). It is unclear from the paper whether all baselines were provided with the same computational budget for the experiments, and, if not, whether the proposed approach would remain competitive when the same compute budget is allocated to all baselines.
- (Minor) The related work section does not mention the first Transformer-based approach, Neural Symbolic Regression that Scales. Additionally, it fails to emphasise that these methods typically use synthetic datasets and are generally quite efficient at inference time (as the cost is amortised during training), without requiring more data than other methods (during inference).

**Questions:**

- In Figure 2, DSR achieves a lower symbolic complexity than your method (albeit with lower accuracy). Considering that you are also using BIC, I find this surprising. Do you have any insights into why this might be the case?
- What happens when the noise is non-Gaussian? For example, if there is localized noise affecting only a subset of data points or noise sampled from a different distribution, how does your method’s performance deteriorate?
- How does performance and the expression complexity vary with the number of data points and the number of variables?
- Could you conduct an experiment to test your approach against DSR with the same compute budget? I am happy to increase my score if you can demonstrate that your method remains competitive in terms of accuracy and interpretability when the baselines (DSR  is sufficient) are given the same compute budget.

---

### Official Review · Reviewer_sXgo · 2024-11-02

**Soundness:** 3
**Presentation:** 3
**Contribution:** 2
**Rating:** 6
**Confidence:** 3

**Summary:**

The paper introduces a novel approach to symbolic regression using transformers with a new reward function (Bayesian Inference Criterion), and policy gradient. This new method is empirically shown to have strong performance compared to other models in terms of output simplicity, especially when dealing with noisy data. The authors also conducted an ablation study to identify the contributions of each component in their method.

**Strengths:**

This paper details a modification of the transformers architecture (Vaswani 2023), in addition to a new reward function and policy gradient. Transformers were originally used in the context of machine translation but can be applied in an effective way to other problems including symbolic regression as described in this paper.

The authors also incorporate Bayesian Inference Criterion, a reward function which takes into account both data fitness and expression complexity, into a novel risk-seeking policy. This pushes discovered equations towards simple expressions, which greatly improves interpretability while avoiding overfitting to noisy data.

**Weaknesses:**

This paper gives me an impression that it is a combination of existing ideas. Hence the significance of the work may be limited.

Transformers have previously been applied to symbolic regression (“End-to-end symbolic regression with transformers”, Kamienny 2022), in a very similar expression generation method as presented in the paper.  The reviewer would be curious to see a comparison between CADSR and the method presented in Kamienny’s paper.

The risk-seeking policy has also been used before, (“Deep symbolic regression: Recovering mathematical expressions from data via risk-seeking policy gradients”, Peterson 2019).  The reviewer would be interested to see comparisons between reward functions other than that of the original paper and BIC.

Controlling the complexity of the equations found in symbolic regression also has been studied in previous works.

**Questions:**

Why is DGSR not used as a comparison base?

A limitation on the method presented in the paper is the training inefficiency – how would the increased model size for a DGSR approach affect the efficiency compared to that of CADSR?

---

### Official Review · Reviewer_SArE · 2024-11-02

**Soundness:** 3
**Presentation:** 3
**Contribution:** 2
**Rating:** 5
**Confidence:** 4

**Summary:**

The paper proposes CADSR, an enhancement to DSR model (Petersen et al., 2019) with three main modifications: (1) replacing RNN with Transformer architecture, (2) using breadth-first search instead of preorder traversal for expression representation, and (3) introducing BIC reward to balance fitting accuracy and expression complexity. The approach is evaluated on the SRBench dataset showing competitive performance in the accuracy-complexity trade-off.

**Strengths:**

* Strong empirical results on SRBench benchmark
* Clear ablation analysis demonstrating the contribution of different component compared to prior works
* Theoretical analysis of the robust risk-seeking policy

**Weaknesses:**

**Major Concerns:**
* I'm wondering how this work differ from current works in the literature. It seems that using transformer architecture DSR and complexity-aware rewards are already explored in recent papers (Landajuela et al., 2022; Shojaee et al., 2023; Sun et al., 2023).

* Figure 3 shows CADSR favoring complexity over accuracy compared to TPSR on the pareto front, while Figures 4 and 6 show the opposite (TPSR favoring complexity over accuracy). This raises question for me about the impact of TPSR's exponential complexity term versus BIC's linear term, especially given the paper's focus on complexity-aware DSR.

* What's the motivation for a new pre-training model (Sec 5.3)? Can't we combine CADSR's new components with the existing pre-trained transformer models (example: Biggio et al., 2021; Kamienny et al., 2022)?

* Why are experiments in section 5.2 and  5.3 limited to few selected datasets (5 for ablation, different ones for pre-training)? What's the selection criteria for these examples? Given the high variance and close performance of model variants, I would suggest authors to provide mean/median result statistics across all 119 Feynman problems for a more reliable comparison.

**Minor Comments:**
* Figure 2's R² metrics are too close to meaningfully compare baselines. I would suggest using accuracy metrics with thresholds instead  (R²>0.99 or R²>0.999).
* Minor Typos: "sever" should be "serve" (line 200), unclear and duplicate reference to "(4)" (line 430-431).

**Questions:**

provided in the weaknesses section

---

### Official Review · Reviewer_xqv1 · 2024-11-06

**Soundness:** 3
**Presentation:** 3
**Contribution:** 3
**Rating:** 5
**Confidence:** 4

**Summary:**

This paper presents CADSR, a novel approach to symbolic regression, addressing gradient vanishing and overfitting limitations of models like DSR by replacing RNN with Transformer architecture. CADSR improves the generation of interpretable mathematical expressions, balancing complexity and data fitness through Transformer and BFS. The authors use the Bayesian Information Criterion as the reward function, optimizing simplicity, interpretability, and accuracy. A modified risk-seeking policy gradient further mitigates gradient vanishing for stable updates. Several benchmark evaluations show CADSR's advantages in interpretability, accuracy, and robustness.

**Strengths:**

1. The paper is well-structured, with clear logic, problem motivation, and detailed descriptions.
2. The authors provide theoretical proofs for the proposed BIC reward and Robust Risk-Seeking Policy.
3. Extensive experiments and analyses are conducted.

**Weaknesses:**

1. The innovation is limited; the primary contribution is applying Transformer architecture to symbolic regression, which lacks novelty. The reward design and strategy gradient adjustments appear as minor training techniques, with limited contribution and impact.
2. Only a subset of examples was used in the ablation experiments, reducing the results’ persuasiveness.
3. There are minor typos, such as "Table ??" and “bigger rewards” in C.1. Fonts in Figures 2 and 3 are too small.

**Questions:**

1. Could the actor module use a decoder-only architecture? How would its performance compare to the current setup?
2. The related work section could better structure the research context before discussing differences and advantages of the proposed approach.

---

### Note · Authors · 2024-11-25

**Comment:**

To whom it may concern,

We are withdrawing our paper from the conference due to the discovery that the quality of the DSR baseline used to validate the performance of CADSR needs to be revised due to different hyperparameters. We are withdrawing the paper for sufficient time to redo the DSR baseline with identical hyperparameters as CADSR to allow for a faithful comparison between methods. We want to thank all the reviewers for their time and work on our paper and will address all the suggestions before our next submission.

Sincerely,
The Authors

**Withdrawal Confirmation:**

I have read and agree with the venue's withdrawal policy on behalf of myself and my co-authors.